# To Drink or Not to Drink? Investigating Alcohol’s Impact on Prostate Cancer Risk

**DOI:** 10.3390/cancers16203453

**Published:** 2024-10-12

**Authors:** Aris Kaltsas, Michael Chrisofos, Evangelos N. Symeonidis, Athanasios Zachariou, Marios Stavropoulos, Zisis Kratiras, Ilias Giannakodimos, Asterios Symeonidis, Fotios Dimitriadis, Nikolaos Sofikitis

**Affiliations:** 1Third Department of Urology, Attikon University Hospital, School of Medicine, National and Kapodistrian University of Athens, 12462 Athens, Greece; ares-kaltsas@hotmail.com (A.K.); mxchris@yahoo.com (M.C.); stamarios@yahoo.gr (M.S.); zkratiras@gmail.com (Z.K.); iliasgiannakodimos@gmail.com (I.G.); 2Department of Urology II, European Interbalkan Medical Center, 55535 Thessaloniki, Greece; evansimeonidis@gmail.com; 3Department of Urology, Faculty of Medicine, School of Health Sciences, University of Ioannina, 45110 Ioannina, Greece; zahariou@otenet.gr; 4Department of Urology, Faculty of Medicine, School of Health Sciences, Aristotle University of Thessaloniki, 54124 Thessaloniki, Greece; symeaste@gmail.com (A.S.); helabio@yahoo.gr (F.D.)

**Keywords:** prostate cancer, alcohol consumption, PSA levels, carcinogenesis, chronic alcohol consumption, cancer risk factors, aggressive prostate cancer

## Abstract

**Simple Summary:**

Prostate cancer is one of the most common cancers in men worldwide, and its causes are influenced by both genetic and lifestyle factors. One lifestyle factor that has shown mixed evidence is alcohol consumption. While some studies suggest that heavy drinking increases prostate cancer risk, others show little to no connection. This research aims to clarify the relationship between alcohol consumption and prostate cancer by reviewing and analyzing global studies. By exploring how different types of alcohol and drinking patterns may influence cancer risk, the goal is to provide more precise guidance for healthcare providers and patients. These findings can help inform public health recommendations and future research on cancer prevention strategies, especially for higher-risk populations.

**Abstract:**

Background/Objectives: Prostate cancer (PCa) is a significant global health issue. The relationship between alcohol consumption and PCa risk has been the subject of extensive research, yet findings remain inconsistent. This review aims to clarify the association between alcohol intake and PCa risk, its aggressiveness, and the potential metabolic pathways involved in PCa onset. Methods: A comprehensive literature search was conducted across multiple databases, including PubMed and MEDLINE, focusing on epidemiological studies, meta-analyses, cohort studies, and case–control studies. Studies evaluating alcohol consumption, prostate-specific antigen (PSA) levels, and PCa risk were included. The review also explored the roles of alcohol dehydrogenase (ADH) and aldehyde dehydrogenase (ALDH) in alcohol metabolism. Results: The analysis reveals a complex relationship between alcohol consumption and PCa. Heavy alcohol intake is associated with an increased risk of PCa, particularly more aggressive forms, and higher mortality rates. However, studies also show weak or no association between moderate alcohol consumption and PCa. The variability in findings may be attributed to differences in alcohol types, regional factors, and study methodologies. Conclusions: The link between alcohol consumption and PCa risk is multifaceted. While heavy drinking appears to increase the risk of aggressive PCa, the overall relationship remains unclear. Further research is needed to better understand these associations and inform public health recommendations and cancer prevention strategies.

## 1. Introduction

Prostate cancer (PCa) is one of the most commonly diagnosed cancers in men worldwide, with its incidence being particularly high in developed nations [1]. According to recent global cancer statistics, approximately 1.4 million men were diagnosed with PCa in 2020, and the disease accounted for around 375,000 deaths globally [2]. In Europe, PCa is the most frequently diagnosed cancer among men and the third leading cause of cancer-related mortality [3]. These statistics highlight the significant public health impact of prostate cancer, underscoring the importance of research into prevention, early detection, and more effective treatment strategies [4].

In addition to incidence rates, autopsy studies have shown that the prevalence of prostate cancer increases with age. The prevalence among men under 30 is 5%, but it escalates with an odds ratio (OR) of 1.7 per decade, reaching 59% in men over the age of 79 [5]. Furthermore, autopsy-detected PCa varies by ethnicity and region, with 83% prevalence observed in white United States males aged 71–80 compared to 41% in Japanese males of the same age group [6]. These numbers emphasize not only the widespread nature of PCa but also its considerable variability across populations.

Beyond the statistics, PCa profoundly affects patients’ quality of life and imposes emotional and financial burdens on families, driving ongoing discussions about more effective prevention, early detection, and treatment strategies [7].

The etiology of PCa is complex, involving genetic, environmental, and lifestyle factors. Aging and family history, particularly the presence of first-degree relatives with PCa, are well-established risk factors [8,9]. Additionally, the marked variation in PCa incidence across different ethnic groups points to the significant role of environmental and lifestyle influences that interact with genetic predispositions [10,11].

One lifestyle factor that has drawn considerable attention in PCa research is alcohol consumption. While excessive alcohol intake is a well-known risk factor for several cancers, including liver and breast cancer [12,13,14], its relationship with PCa remains unclear and requires further exploration [15]. Studies on this subject, including cohort studies and meta-analyses, have produced mixed results: some research suggests that heavy alcohol consumption may increase the risk of PCa. In contrast, other studies report no association or even a potential protective effect, particularly with moderate wine consumption due to its polyphenol content [16,17,18].

Alcohol’s influence on hormonal regulation may drive these associations, particularly its impact on testosterone levels, as well as inflammation and oxidative stress, all of which are implicated in prostate carcinogenesis [19,20]. Alcohol’s ability to increase reactive oxygen species (ROS) and disrupt hormonal balance may promote the development of PCa. Understanding how these mechanisms specifically affect the prostate is essential for clarifying the alcohol–PCa relationship [21,22].

The challenge in disentangling alcohol’s impact on PCa is further compounded by the influence of lifestyle factors such as diet, tobacco use, and concurrent treatments, which may obscure or exaggerate the actual risk [23]. Additionally, the type of alcohol consumed, whether beer, wine, or liquor, may have varying effects on prostate health, possibly due to differences in metabolic processing and the presence of protective or harmful compounds [16].

Given the contradictory evidence surrounding alcohol consumption and PCa, this review aims to clarify the epidemiological and metabolic links between the two. This comprehensive analysis not only seeks to provide more precise insights into how different patterns of alcohol consumption influence PCa risk and progression but also aims to inform future research and refine clinical guidelines. Clarifying these associations could lead to more effective clinical practices, such as tailored screening protocols or lifestyle advice for patients at different levels of risk, particularly in high-risk populations.

## 2. Materials and Methods

### 2.1. Search Strategy

A comprehensive literature search was conducted using the PubMed and MEDLINE databases, focusing on articles published in English. The search included studies from the inception up to June 2024. Keywords and search phrases included “alcohol consumption AND prostate cancer”, “alcohol intake AND PCa risk”, “drinking habits AND prostate malignancy”, “alcohol types AND PCa correlation”, and “epidemiological studies AND alcohol AND prostate cancer”. This strategy ensured a broad collection of studies relevant to both alcohol consumption and PCa risk or progression.

### 2.2. Inclusion Criteria

The included studies were peer-reviewed epidemiological studies, meta-analyses, cohort studies, and case–control studies that investigated the association between alcohol consumption and PCa risk, aggressiveness, mortality, or related metabolic pathways. Studies were required to involve adult male participants aged 18 years and older and to provide clear definitions of alcohol consumption levels and prostate cancer outcomes. Only studies published in English and available in full-text form were considered.

### 2.3. Exclusion Criteria

Exclusion criteria were applied to omit studies that could introduce bias or those not directly relevant to our research objectives. We excluded reviews, editorials, letters, and commentaries without original data, as well as animal or in vitro studies. Studies lacking clear definitions of alcohol consumption levels or not available in full-text form were also excluded.

### 2.4. Data Extraction

Data extraction was performed independently by two reviewers (A.K. and E.N.S.) using a standardized data extraction form to ensure consistency and reduce the potential for errors. Extracted data included study characteristics (author(s), publication year, country, study design, sample size), participant characteristics (age range, ethnicity, health status), exposure assessment details (alcohol consumption patterns such as frequency, quantity, type of alcohol, duration of alcohol use), outcome measures related to prostate cancer (incidence, aggressiveness indicated by Gleason score, mortality rates, PSA levels), and information on confounding factors (adjustments made for variables such as age, family history, smoking status, diet, and hormonal therapy). Any discrepancies between the reviewers were resolved through discussion or, if necessary, consultation with a third reviewer (N.S.) to reach a consensus.

### 2.5. Addressing Biases

Attention was paid to minimizing potential biases, particularly misclassification. Studies that reported positive associations with prostate cancer tended to control for more confounding factors such as age, family history, and lifestyle, including diet and tobacco use, while those reporting no associations often lacked this rigor. For instance, former drinkers were carefully distinguished from current drinkers to prevent misclassification in the assessment of alcohol consumption patterns. Further, the failure to distinguish between moderate and heavy drinkers or consider lifetime alcohol consumption may have led to inconsistent results across studies. Confounding factors, such as diet, tobacco use, and hormone therapy, were also accounted for to ensure a more accurate understanding of the relationship between alcohol consumption and PCa. These factors were considered during the data extraction and analysis processes to improve the validity of the findings.

## 3. Alcohol Consumption and Risk of Prostate Cancer

Understanding the complex interplay between alcohol consumption and PCa risk has been the focus of extensive epidemiological research. This section explores the varying conclusions reached by different studies, with some indicating a clear association between alcohol intake and PCa risk, while others show no significant relationship. The discrepancies in these findings are likely attributable to variations in study design, population characteristics, and measurement of alcohol consumption patterns.

### 3.1. Positive Correlation between Alcohol and PCa Risk

A growing body of global evidence suggests that heavy alcohol consumption is significantly associated with an increased risk of developing PCa. Numerous global studies report higher PCa incidence among individuals with a history of alcoholism, with some identifying a clear positive correlation between alcohol intake and age-standardized PCa incidence, prevalence, and mortality rates using linear regression analysis [24,25,26,27,28,29,30]. A landmark meta-analysis up to July 1998 demonstrated a robust connection between heavy alcohol consumption and PCa risk, though moderate intake did not exhibit the same statistically significant relationship [31].

Importantly, studies have consistently suggested a dose–response relationship, meaning that as alcohol consumption increases, so does the risk of developing PCa. This pattern was confirmed by a comprehensive meta-analysis of 340 studies, which revealed a substantial increase in PCa risk at varying levels of alcohol consumption [32]. Additional evidence from a South Korean population study corroborated this, showing that higher levels of alcohol consumption were linked to proportionally greater PCa risk [33].

The geographical breadth of research also adds weight to these findings. Studies conducted in Japan, Brazil, and Algeria have consistently shown elevated PCa risk in individuals with frequent alcohol consumption, highlighting the global relevance of this relationship [34,35,36]. Furthermore, evidence from the United States indicates that even daily alcohol consumption, when not classified as excessive, is associated with a 25% increase in the risk of non-advanced PCa [37,38]. Research conducted by the NCI Breast and Prostate Cancer Cohort Consortium (BPC3) found a similar increase in risk when daily alcohol consumption exceeded 30 g [39].

The impact of different types of alcohol on PCa risk remains a subject of debate. Some research has suggested a potential protective effect of certain compounds found in wine, notably polyphenols, which are believed to have antioxidant properties [40,41]. However, the findings are mixed, with other studies linking red and white wine consumption to varying degrees of PCa risk depending on intake levels and individual characteristics [42,43]. Similarly, research shows that the dangers posed by hard liquor and beer may be higher due to carcinogenic compounds such as N-nitroso found in beer [44,45,46,47]. These variations emphasize the importance of further research to clarify the role that different alcoholic beverages play in PCa development.

### 3.2. Weak or No Correlation between Alcohol and PCa Risk

In contrast to studies suggesting a positive correlation between alcohol and PCa, some investigations report no significant relationship between alcohol intake and PCa risk. For instance, the Nutrition Canada Survey (1970–1972), which followed 145 PCa cases, found no significant link between overall alcohol consumption and subsequent PCa development. However, minimal wine consumption was associated with a slight increase in risk [48].

Similarly, studies conducted in Iowa and Hawaii failed to establish a significant connection between general alcohol intake or specific types of alcohol and PCa risk [49,50]. Large cohort studies from Norway and South Korea, involving over 20,000 participants, also did not observe a meaningful relationship between alcohol consumption and PCa risk, even among heavy drinkers [51,52]. These findings are supported by research from the UK, where data from nearly 9000 adults, cross-referenced with cancer registry records, did not demonstrate a significant association between alcohol consumption and PCa risk [53].

Methodological differences between studies could explain some of the discrepancies in the results. Variations in how alcohol consumption is measured, as well as differences in population demographics, genetic predispositions, and lifestyle factors, complicate the ability to isolate the true effect of alcohol on PCa. Furthermore, studies that report no correlation often lack data on lifetime alcohol intake or detailed drinking patterns, making it challenging to draw definitive conclusions [54,55,56,57,58,59,60].

It is also possible that regional differences in alcohol consumption patterns and genetic susceptibility to PCa contribute to the observed variations in these studies. For example, some populations may have different genetic predispositions related to alcohol metabolism, which could influence how alcohol affects cancer risk in those populations. Further research that accounts for these genetic factors is necessary to fully understand the relationship [60,61,62,63].

### 3.3. Summary of Alcohol’s Role in PCa Risk

The relationship between alcohol consumption and PCa risk remains complex. Numerous meta-analyses have investigated this association, yielding varying results. While a substantial body of evidence supports a positive correlation between excessive alcohol intake and increased PCa risk, particularly in the form of a dose–response relationship, other studies present conflicting results, especially concerning moderate or light drinking. Table 1. summarizes key findings from several meta-analyses examining the association between alcohol consumption and PCa risk.

Many of the inconsistencies in the findings on the association between alcohol consumption and prostate cancer can be attributed to variations in study design and potential biases. Studies that failed to account for confounding factors such as family history, concurrent medications, or lifestyle habits often reported no association between alcohol consumption and prostate cancer risk. Conversely, studies with rigorous controls, particularly those that differentiated between former and current drinkers and adjusted for confounders, were more likely to report a positive association, particularly with heavy drinking. The heterogeneity in alcohol measurement—whether in terms of frequency, quantity, or type of alcohol—also significantly affected the reported outcomes.

## 4. Metabolic and Biological Implications of Alcohol in Prostate Cancer

Research indicates that alcohol consumption plays a multifaceted role in PCa development, mainly due to its effects on metabolic and genetic pathways. There is a growing body of evidence linking genetic susceptibility to alcohol’s carcinogenic effects, particularly through variations in genes associated with alcohol metabolism. For instance, in a study on cancer-related metabolites, alcohol consumption ranked as the sixth most influential factor in metabolite concentration variability among PCa patients, accounting for 1.1% of the variation [67].

### 4.1. Alcohol Metabolism and Its Role in Carcinogenesis

Alcohol metabolism in the body primarily involves two critical enzymes: alcohol dehydrogenase (ADH) and aldehyde dehydrogenase (ALDH). When ethanol (EtOH) is consumed, ADH catalyzes the oxidation of ethanol to acetaldehyde (ACH), a highly reactive and toxic metabolite. Acetaldehyde is further metabolized to acetate by ALDH, which is less harmful and eventually excreted [68,69]. However, variations in the activity of these enzymes, often due to genetic polymorphisms, can lead to differences in acetaldehyde accumulation, impacting cancer risk [70].

Individuals with genetic mutations in ADH or ALDH, particularly in populations such as East Asians with ADH1B and ALDH2 mutations, exhibit a slower breakdown of acetaldehyde. This leads to prolonged exposure to the carcinogenic effects of acetaldehyde, contributing to a higher risk of alcohol-related cancers [71,72,73]. Notably, prostate cancer patients often demonstrate elevated total ADH activity, which results in greater acetaldehyde production. This elevation in ADH activity has been observed both systemically, in serum and in prostate tissue itself, highlighting its potential role in the local and systemic metabolic pathways that contribute to PCa carcinogenesis [74,75]. Acetaldehyde is a known carcinogen, and its accumulation in prostate tissue can promote DNA damage by forming DNA adducts and inhibiting DNA repair mechanisms [76,77]. Acetaldehyde disrupts the function of O6-methylguanine-DNA methyltransferase, a critical enzyme for repairing DNA alkylation damage caused by carcinogens [78].

The role of ALDH in prostate cancer is paradoxical. On the one hand, ALDH assists in detoxifying acetaldehyde and synthesizing retinoic acid, which inhibits tumor growth [79,80]. On the other hand, the elevated activity of specific ALDH isoforms, particularly ALDH1A1, has been associated with more aggressive forms of prostate cancer, reflected in higher Gleason scores and poorer prognosis [81,82,83,84,85,86]. This enzyme is highly expressed in cancer stem cells, which are responsible for tumor initiation, maintenance, and resistance to conventional therapies [87,88,89].

ALDH1A1 contributes to cancer progression by promoting epithelial-to-mesenchymal transition (EMT), a process that allows cancer cells to become more invasive and metastatic [90]. Additionally, ALDH1A1 regulates the synthesis of retinoic acid, which can influence gene expression patterns involved in tumor growth [91]. Therefore, although ALDH enzymes generally aid in detoxifying carcinogens like acetaldehyde, ALDH1A1 paradoxically supports aggressive tumor behavior in prostate cancer.

### 4.2. Acetaldehyde and Oxidative Stress in Prostate Cancer

Acetaldehyde is a potent carcinogen that can cause significant cellular damage through its interactions with DNA. The formation of DNA adducts by acetaldehyde can result in mutations, chromosomal rearrangements, and genomic instability, all of which are hallmarks of cancer. In prostate cells, these mutations may accumulate over time, leading to the initiation of cancerous growths. Acetaldehyde’s inhibition of DNA repair enzymes further exacerbates this damage, allowing mutated cells to proliferate unchecked [92,93].

Additionally, acetaldehyde induces oxidative stress by generating ROS during its metabolism. Oxidative stress damages cellular components, including lipids, proteins, and DNA, contributing to carcinogenesis [94]. ROS are particularly damaging to prostate cells, as they disrupt cellular homeostasis and promote inflammatory pathways that facilitate tumor growth and metastasis [22]. Furthermore, acetaldehyde alters folate metabolism, which is essential for DNA synthesis and repair. By depleting folate reserves, acetaldehyde indirectly increases the likelihood of DNA mutations, adding another layer to its carcinogenic potential [95].

Chronic alcohol consumption can activate an alternative ethanol metabolism pathway through the cytochrome P450 2E1 (CYP2E1) enzyme. This pathway generates additional ROS, leading to oxidative stress and DNA damage—critical factors in prostate cancer progression [96].

Further studies by Castro and colleagues have identified other pathways contributing to the carcinogenic process. For instance, cytosolic xanthine oxidase in the prostate can convert acetaldehyde into radicals, further amplifying oxidative stress. Additionally, ethanol can be activated into acetaldehyde and 1-hydroxyethyl in prostate microsomes through oxidases and peroxidases, exacerbating DNA damage [97,98]. In prostate tumor tissues, elevated enzymatic activities and increased markers of lipid peroxidation have been detected compared to healthy controls, reinforcing the role of oxidative damage in cancer development [99].

### 4.3. Alcohol’s Impact on Testosterone and Hormonal Balance

Alcohol consumption has significant effects on hormonal regulation, particularly testosterone levels, which are intricately linked to prostate health and the development of PCa. Elevated circulating testosterone is associated with an increased susceptibility to PCa, and lifestyle factors, including alcohol consumption, can influence testosterone levels and related behavioral patterns [100,101,102,103]. For instance, individuals who consume hard liquor often exhibit behaviors such as early first intercourse and a higher number of sexual partners [44]. This may be linked to alcohol’s impact on testosterone levels, which not only increases sexual behaviors but could also elevate PCa risk by promoting hormonal changes associated with carcinogenesis [104].

Testosterone in circulation exists in two main forms: free testosterone, which is biologically active, and testosterone bound to serum proteins, such as sex hormone-binding globulin (SHBG) and albumin. Bound testosterone is largely inactive, while free testosterone can readily bind to androgen receptors in the prostate, influencing cancer development. This distinction is crucial, as free testosterone is more strongly associated with PCa risk and progression than total testosterone. In this context, reduced SHBG levels, often seen in chronic alcohol users, can increase the proportion of free testosterone, which may enhance PCa progression [105,106,107,108,109].

The relationship between alcohol and testosterone is complex and dose-dependent. Research shows that acute alcohol intake can lead to a short-term increase in plasma testosterone levels, potentially raising PCa risk due to elevated androgenic activity [110]. This temporary rise in free testosterone may contribute to heightened androgen receptor signaling, which can promote carcinogenic pathways in prostate tissue [111].

However, chronic alcohol consumption typically results in a significant decrease in testosterone levels due to increased metabolic clearance and liver activity [112]. Chronic alcohol use has also been shown to reduce SHBG levels, which paradoxically increases the amount of free testosterone despite the decrease in total testosterone [57]. This creates a scenario where lower SHBG levels elevate the fraction of bioavailable testosterone, which can continue to drive PCa progression [104,113].

The duality of alcohol’s effects on testosterone—raising free testosterone acutely while lowering total testosterone chronically—highlights the complexity of its influence on PCa risk. Despite the decline in overall testosterone levels with long-term alcohol consumption, the increased proportion of free testosterone due to reduced SHBG levels may still contribute to PCa progression. This complex interaction is observed in human studies, where inconsistencies arise based on alcohol consumption patterns [110,114]. In a cross-sectional study from China, former drinkers were found to have higher levels of luteinizing hormone and free testosterone compared to non-drinkers [115]. In contrast, men who consumed large amounts of alcohol showed lower testosterone levels, indicating the dose-dependent impact of alcohol on testosterone concentrations [114].

Chronic alcohol consumption is also linked to endocrine dysfunction, such as pseudo-Cushing’s syndrome and disruptions to the hypothalamic–pituitary–adrenal (HPA) axis, further exacerbating hormonal imbalances that may contribute to PCa risk [116,117,118].

Animal studies corroborate these findings, demonstrating that chronic alcohol consumption reduces serum testosterone levels and alters the hypothalamic–pituitary axis, leading to reduced prostate size and testicular atrophy. These changes reflect alcohol’s broader impact on the male reproductive system. Ethanol administration in rat models has been shown to reduce serum testosterone levels and disrupt the hypothalamic–pituitary axis, coupled with weight reductions in the prostate and reproductive organs. These effects are further complicated by genetic differences in alcohol metabolism and study methodologies, which may explain some of the discrepancies observed in the relationship between alcohol, testosterone, and PCa risk [119,120,121,122,123].

In summary, alcohol’s influence on testosterone and SHBG levels, as well as its broader hormonal and metabolic effects, contributes to a complex relationship with PCa risk. While acute alcohol consumption may transiently raise testosterone levels, chronic consumption lowers total testosterone but may elevate free testosterone due to decreased SHBG, enhancing the risk of PCa progression. This highlights the multifaceted ways in which alcohol consumption impacts the endocrine system and PCa development, requiring further investigation into these mechanisms.

### 4.4. Alcohol’s Effect on PSA Levels and Implications for PCa Detection

Alcohol consumption has been shown to influence prostate-specific antigen (PSA) levels, which are critical markers in the detection and diagnosis of PCa. PSA is a protein produced by both normal and malignant prostate cells, and elevated PSA levels often prompt further investigation for possible PCa [124]. However, several studies suggest that alcohol consumption may lower PSA levels, potentially complicating the effectiveness of PSA testing and delaying early PCa detection [125].

A UK-based study investigated the relationship between self-reported alcohol intake and PSA levels among 2400 men with PCa and 12,700 age-matched controls. The results indicated that heavy drinking, defined as consuming at least five drinks per day/week, was associated with lower PSA levels, which could hinder early PCa detection. Interestingly, despite the reduced PSA levels, these heavy drinkers showed a slightly higher risk of developing aggressive forms of PCa [126]. Similarly, a large study involving 212,039 male participants from the UK Biobank found a correlation between reduced alcohol intake and a lower likelihood of undergoing PSA testing, suggesting that lower alcohol consumption may be linked to fewer PSA screenings [127].

Research from New Zealand further supports these findings. A study of 1,031 men identified a strong inverse correlation between beer consumption and both total and free serum PSA levels. This inverse relationship implies that beer, in particular, may lower PSA levels and potentially delay the detection of PCa [128].

In patients diagnosed with PCa who consume both alcohol and tobacco, PSA levels tend to be elevated compared to non-consumers. This suggests that the combined effects of alcohol and tobacco may further complicate PSA test results and hinder early PCa detection [129]. Additionally, temporary lifestyle changes such as abstaining from alcohol, coffee, and spicy foods have been found to reduce PSA levels, further underscoring the influence of lifestyle factors on PSA measurements [130].

Furthermore, studies from Croatia and South Korea have linked increased weekly alcohol consumption to lower PSA levels in individuals without known prostate health issues. This association reinforces the need to consider alcohol consumption when interpreting PSA levels in both healthy individuals and those at risk for PCa [130,131].

The potential for alcohol to reduce PSA levels has significant implications for PCa detection. Lower PSA levels might mask the presence of prostate cancer, particularly in heavy drinkers, leading to delayed diagnosis and potentially more advanced disease at the time of detection. Therefore, it is recommended that clinicians account for patients’ alcohol consumption habits when interpreting PSA results. Inquiring about alcohol intake could improve the accuracy of PSA-based assessments and ensure earlier detection of PCa in at-risk individuals [126].

In conclusion, alcohol consumption has a measurable effect on PSA levels, with studies consistently showing a reduction in PSA levels among moderate to heavy drinkers. While the exact mechanisms remain to be fully elucidated, this relationship complicates the use of PSA as a reliable diagnostic tool for PCa, highlighting the importance of integrating lifestyle factors into clinical assessments.

### 4.5. Summary of Metabolic Impacts

The metabolic and hormonal implications of alcohol consumption in prostate cancer are complex and multifaceted. Acetaldehyde, a byproduct of alcohol metabolism, plays a central role in inducing DNA damage, oxidative stress, and disruptions to DNA repair mechanisms, all of which contribute to carcinogenesis. Additionally, the hormonal effects of alcohol on testosterone and PSA levels highlight its potential to influence prostate cancer risk. While the relationship between alcohol and PCa remains nuanced, with both dose-dependent and beverage-specific effects, the evidence underscores the importance of understanding how alcohol consumption can impact prostate health at multiple levels. Further research is necessary to elucidate the precise mechanisms by which alcohol influences prostate carcinogenesis and to develop targeted prevention strategies for those at risk. The following Figure 1 illustrates the dual role that alcohol plays in prostate carcinogenesis and tumor progression, emphasizing how EtOH and its metabolites can accelerate both cancer initiation and progression through direct and indirect mechanisms.

## 5. Alcohol Consumption and Prostate Cancer Aggressiveness

### 5.1. Alcohol’s Link to Aggressive Forms of PCa

Chronic alcohol consumption has been linked to the development of more aggressive forms of PCa, including those with a higher Gleason score and increased metastatic potential. Several studies have established this link between long-term alcohol intake and aggressive PCa outcomes, showing that men who consume alcohol regularly, especially in large quantities, face a higher risk of developing advanced-stage prostate cancer [126,132,133].

A study conducted in the U.S. found a correlation between high alcohol consumption and an increased likelihood of developing aggressive prostate tumors, characterized by higher Gleason scores and a greater propensity for metastasis. The study also demonstrated that men who consumed beer or hard liquor more frequently were more likely to have aggressive forms of the disease, with beer being the most strongly associated with advanced prostate cancer [134]. Similarly, an Australian cohort study identified that consuming alcohol, particularly beer, five or more days a week significantly raised the risk of developing prostate cancer, with Gleason scores of eight or higher [135].

In Japan, research demonstrated that as weekly alcohol consumption increased (0–149 g/week, 150–299 g/week, and ≥300 g/week), so did the risk of aggressive prostate cancer, suggesting a dose–response relationship between alcohol intake and cancer severity [136]. This aligns with findings from military veterans in the U.S., where those who consumed more alcohol during their youth had a higher risk of developing aggressive prostate cancer later in life [137].

### 5.2. Alcohol Consumption and PCa Mortality Data

The correlation between alcohol consumption and prostate cancer mortality has also been the subject of multiple studies. Data from a Canadian study with a 19-year follow-up period indicated that prostate cancer patients who consumed more than eight drinks per week had significantly higher mortality rates than those who either reduced their alcohol intake or stopped drinking post-diagnosis [138].

In the U.S., regions with stricter alcohol regulations, such as those limiting the availability of alcohol or raising taxes on alcoholic beverages, have reported lower prostate cancer mortality rates. This association between alcohol consumption and mortality is supported by findings from an analysis of military veterans and population-based studies, indicating that reducing alcohol intake to less than one drink per day could decrease prostate cancer mortality by a significant margin [139]. Similarly, Japanese studies have reported that men who consume alcohol daily are at a 2.5-times-higher risk of dying from prostate cancer compared to non-drinkers [140].

Global data corroborates these findings, as countries with higher alcohol consumption tend to exhibit higher prostate cancer mortality rates. For instance, countries such as Bhutan and Nepal, where alcohol consumption is minimal, report some of the lowest prostate cancer mortality rates worldwide [141].

Mendelian randomization studies from the PRACTICAL Consortium further strengthened the evidence linking alcohol consumption with prostate cancer aggressiveness and mortality. These studies employed genetic markers such as single nucleotide polymorphisms (SNPs) to explore the impact of alcohol on prostate cancer progression. Notably, an SNP in the ALDH1B1 gene was found to be associated with increased mortality in low-grade prostate cancer patients, indicating that genetic factors may influence how alcohol affects cancer progression [70]. Additionally, high alcohol intake was linked to more aggressive forms of prostate cancer in genetic subgroups related to cell cycle regulation, adhesion, and tumor suppression [142].

Moreover, research on genetic variations in genes like TNF-α and IL-10 has found associations with both prostate cancer aggressiveness and alcohol consumption. These genes, which are involved in immune response and inflammation, may interact with alcohol to enhance the risk of developing aggressive prostate cancer, as demonstrated in studies on the Indian population [143,144,145].

In summary, chronic alcohol consumption is not only linked to an increased risk of developing aggressive forms of prostate cancer but is also associated with higher mortality rates. The relationship between alcohol intake and prostate cancer progression appears to be influenced by both the quantity of alcohol consumed and genetic factors that affect alcohol metabolism and inflammatory responses. Reducing alcohol intake, particularly in high-risk populations, may offer a potential strategy to lower both the incidence of aggressive prostate cancer and the associated mortality rates.

### 5.3. Alcohol’s Role in Androgen Deprivation Therapy (ADT)

Androgen deprivation therapy (ADT) is a widely used treatment for prostate cancer, particularly in castration-resistant prostate cancer (CRPC) patients [146]. However, alcohol consumption can influence the outcomes of ADT, even when consumed in moderate amounts. Studies have shown that Leuprolide, an LHRH agonist used in ADT, may mitigate alcohol withdrawal symptoms but can also slow the progression towards alcohol dependence with extended use [147].

Chronic alcohol exposure has been shown to enhance anxiety-like behavior during alcohol withdrawal in preclinical models. Leuprolide, when used as part of ADT, has been found to reduce this withdrawal-induced behavior, suggesting potential interactions between alcohol and hormone therapy [148]. Additionally, alcohol consumption may exacerbate common side effects of ADT, such as osteoporosis [149,150]. Heavy alcohol use, combined with prolonged ADT, increases the risk of osteoporosis and related fractures, particularly in patients with low serum 25-hydroxyvitamin D levels [151,152].

Furthermore, standard ADT side effects like hot flashes, nausea, and fatigue may be worsened by concurrent alcohol consumption [153]. Given the importance of ADT in managing advanced prostate cancer, understanding the role of alcohol in influencing treatment side effects and outcomes is crucial for improving patient management strategies [154].

## 6. Study Limitations and Future Research Directions

### 6.1. Study Limitations

Despite the growing body of research on the relationship between alcohol consumption and prostate cancer (PCa), several limitations persist across the literature. One fundamental limitation is the reliance on observational studies prone to bias and confounding factors. Many studies do not adequately control for lifestyle factors such as diet, smoking, and exercise, which may influence both alcohol consumption patterns and PCa risk. Additionally, there is significant variability in the measurement of alcohol intake, with studies often lacking standardized definitions for “moderate” or “excessive” consumption. This makes it difficult to compare findings across studies and draw definitive conclusions about the dose–response relationship between alcohol and PCa.

Another limitation is the frequent misclassification of former drinkers as non-drinkers in study designs, which can skew results and mask potential risks associated with alcohol cessation or history. Moreover, many of the studies lack long-term follow-up, which is crucial for understanding the chronic effects of alcohol on PCa development and progression. Genetic predispositions that may affect alcohol metabolism are often not accounted for, limiting our understanding of how individual variations in genes such as ADH and ALDH contribute to PCa risk.

### 6.2. Suggestions and Avenues for Future Research

To address these limitations, future research should prioritize the use of long-term, prospective cohort studies with more rigorous control of confounding variables. Standardizing the definition of alcohol consumption levels—such as distinguishing between moderate, heavy, and binge drinking—would improve comparability between studies. Furthermore, future studies should focus on the effects of different types of alcohol (beer, wine, spirits) on PCa risk and the interaction between alcohol and other risk factors like diet, hormonal therapy, and genetic susceptibility.

Research into the genetic factors that influence alcohol metabolism, particularly concerning enzymes like ADH and ALDH, is also needed. Such studies could clarify why some individuals may be more vulnerable to alcohol-induced carcinogenesis than others. Additionally, given the mixed findings on alcohol’s role in prostate cancer aggressiveness and mortality, further research should examine how alcohol interacts with specific treatments like androgen deprivation therapy (ADT) and how it may impact treatment outcomes.

Lastly, more comprehensive studies are needed to explore the relationship between alcohol consumption, PSA levels, and prostate cancer detection. Understanding whether alcohol consumption suppresses PSA levels and delays diagnosis could lead to more personalized PSA screening recommendations for men with varying alcohol consumption patterns. These insights could enhance public health guidelines, allowing for better-informed recommendations on alcohol consumption and prostate cancer prevention.

## 7. Conclusions

The relationship between alcohol consumption and PCa is multifaceted, with evidence suggesting that excessive alcohol intake is linked to increased PCa risk, aggressiveness, and mortality. Conversely, moderate alcohol consumption has shown more variable outcomes, potentially influenced by the type of alcohol, genetic factors, and metabolism pathways involving ADH and ALDH enzymes. Alcohol’s impact on PSA levels can complicate early PCa detection. Methodological inconsistencies among studies, including variability in alcohol measurement and lack of control for confounders, limit the current understanding. More comprehensive studies are needed to fully understand the implications of alcohol consumption on PCa. Public health strategies should incorporate these nuanced findings to offer more personalized prevention and management approaches.

## Figures and Tables

**Figure 1 cancers-16-03453-f001:**
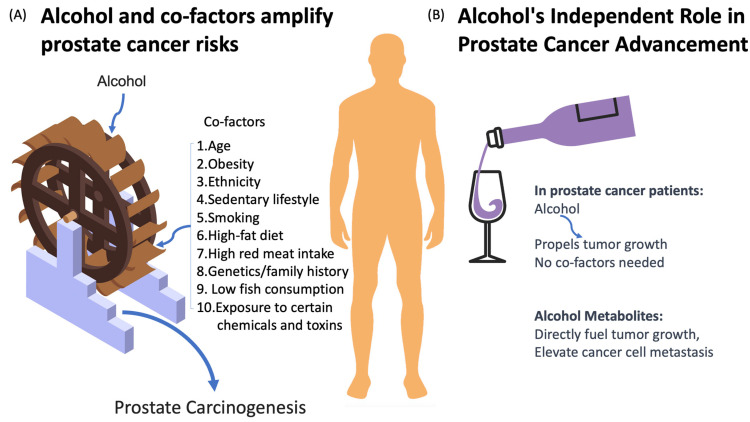
The dual role of alcohol in prostate carcinogenesis and tumor progression. (**A**) Alcohol, especially ethanol (EtOH), accelerates prostate carcinogenesis, with its effects intensified by lifestyle choices, genetic factors, and socio-economic conditions. (**B**) Post-diagnosis, EtOH metabolites alone can amplify tumor growth and metastasis, independent of other cofactors.

**Table 1 cancers-16-03453-t001:** Key meta-analyses examining the association between alcohol consumption and prostate cancer risk.

Study	Type	Population	Alcohol Consumption Categories	Relative Risk (RR)	Dose–Response Relationship	Conclusion
Hong et al., 2020 [16]	Meta-analysis	11 cohort studies	Liquor: RR = 1.04 (95% CI, 1.02–1.06), Beer: RR = 1.03 (95% CI, 1.01–1.05 at 14 g/day)	Beer: Non-linear; Liquor: Linear increase	Yes	Positive association for liquor and beer intake with non-aggressive PCa; no significant association for wine.
Zhao et al., 2016 [32]	Meta-analysis	27 studies	Low (>1.3 g < 24 g/day), Medium (25–44 g/day), High (45–64 g/day), Higher (>65 g/day)	Low: 1.08 (*p* < 0.001), Medium: 1.07 (*p* < 0.01), High: 1.14 (*p* < 0.001), Higher: 1.18 (*p* < 0.001)	Yes (*p* < 0.01)	Significant dose–response relationship; increased risk starting at low volume.
Rota et al., 2012 [64]	Meta-analysis	72 studies (50 case–control, 22 cohort), 52,899 PCa cases	Light (≤1 drink/day), Moderate (1–4 drinks/day), Heavy (≥4 drinks/day)	Light: 1.05 (95% CI, 1.02–1.08), Moderate: 1.06 (95% CI, 1.01–1.11), Heavy: 1.08 (95% CI, 0.97–1.20)	No significant trend for heavy drinkers	No evidence of a material association between alcohol drinking and prostate cancer, even at high doses.
Fillmore et al., 2009 [65]	Meta-analysis	35 studies	Light, Moderate, Heavy	Positive association for heavy drinking (OR = 1.16, CI: 1.06–1.26)	Yes, for population case–control studies	Positive linear association between heavy alcohol consumption and prostate cancer, especially in population case–control studies. No significant association in cohort or hospital case–control studies.
Bagnardi et al., 2001 [66]	Meta-analysis	11 studies, 4094 cases	25 g/day, 50 g/day, 100 g/day	25 g/day: 1.05 (95% CI, 1.00–1.08), 50 g/day: 1.09 (95% CI, 1.02–1.17), 100 g/day: 1.19 (95% CI, 1.03–1.37)	Yes	Weak but significant increase in PCa risk for higher levels of alcohol consumption.
Dennis, 1999 [31]	Meta-analysis	33 studies (6 cohort, 27 case–control)	Beer (RR = 1.27), Alcohol (RR = 1.05 for 1 drink/day, 1.21 for 4 drinks/day)	Overall RR = 1.05 (95% CI, 0.91–1.20)	Linear dose–response for increased alcohol consumption	No overall association between alcohol consumption and prostate cancer; some bias towards positive associations in certain categories.

## Data Availability

Not applicable.

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
