# Peer review of "To Drink or Not to Drink? Investigating Alcohol’s Impact on Prostate Cancer Risk"

_cancers, 2024, doi:10.3390/cancers16203453_

Round 1
Reviewer 1 Report
Comments and Suggestions for Authors
The manuscript provides an analysis of available data to explore the potential correlation between alcohol consumption and the risk of prostate cancer (PCa), as well as the relationship between drinking patterns and the aggressiveness of PCa.
Major Issues:
Section 4.3: This section could be written more clearly, as it is currently unclear whether the alcohol-testosterone-PCa axis is truly significant for cancer development. The authors briefly mention this connection in lines 276-282 but do not fully develop the discussion. It would benefit from a more thorough explanation of how alcohol consumption may impact testosterone levels and how this relates to PCa risk and progression.
Line 271: The statement “(…) Elevated circulating testosterone is associated with an increased susceptibility to PCa” requires stronger references. Testosterone in circulation is mostly bound to serum proteins such as SHBG (sex hormone-binding globulin) and albumin. Is there a clear distinction between the correlation of protein-bound versus free (unbound) testosterone with PCa risk? Clarifying this would strengthen the argument. Please provide studies or data that specifically address the role of these different testosterone fractions (bound and free) in PCa development or aggressiveness.
Line 280: The term “bioavailable testosterone” includes both SHBG-bound testosterone and free testosterone. The authors should specify which fraction they are referring to when discussing a reduction in bioavailable testosterone. Clearer distinction between SHBG-bound testosterone and free testosterone would improve the scientific accuracy of the manuscript.
Minor Issues:
Lines 223-224: The statement “(…) prostate cancer patients often demonstrate elevated total ADH activity, which results in greater acetaldehyde production” would benefit from clarification regarding the specific tissues involved. Is this elevation observed in prostate tissue, or is it a systemic feature? Clarifying this would help the reader understand the context better.
Lines 229-235: The role of ALDH1A1 in promoting cancer progression is mentioned but not fully explained. The manuscript would benefit from a more detailed explanation of the molecular mechanism through which ALDH1A1 contributes to carcinogenesis. How does this enzyme specifically promote prostate cancer progression?
Lines 48-49: It would be helpful to provide specific statistics on the incidence of prostate cancer in men. Including concrete numbers would give readers a clearer understanding of the scale of the issue and the relevance of the study.
In summary, while the manuscript presents an interesting analysis, addressing these points will improve the clarity and scientific rigor of the discussion, particularly with regard to the role of testosterone, enzyme activity, and the alcohol-PCa connection.
Author Response
Dear Reviewer 1,
Thank you for your detailed and insightful feedback on our manuscript. We have carefully considered your comments and made the necessary revisions to enhance the clarity and scientific rigor of the discussion. Below, we address each of your major and minor concerns point by point.
Major Issues:
- Section 4.3: Clarifying the alcoholtestosteronePCa axis
Reviewer comment:
“This section could be written more clearly, as it is currently unclear whether the alcoholtestosteronePCa axis is truly significant for cancer development. The authors briefly mention this connection in lines 276282 but do not fully develop the discussion. It would benefit from a more thorough explanation of how alcohol consumption may impact testosterone levels and how this relates to PCa risk and progression.”
Response:
We have expanded Section 4.3 to more clearly explain the relationship between alcohol consumption, testosterone levels, and PCa risk. The revised text now provides a more detailed discussion of the dosedependent effects of alcohol on both total and free testosterone levels, highlighting how acute alcohol intake may transiently increase free testosterone, whereas chronic alcohol consumption leads to a reduction in total testosterone due to increased metabolic clearance. Additionally, we have provided a more thorough analysis of the role of SHBGbound and free testosterone in PCa development and aggressiveness. Relevant studies and references have been added to support this discussion.
- Line 271: Distinguishing between proteinbound and free testosterone
Reviewer comment:
“The statement ‘(…) Elevated circulating testosterone is associated with an increased susceptibility to PCa’ requires stronger references. Testosterone in circulation is mostly bound to serum proteins such as SHBG (sex hormonebinding globulin) and albumin. Is there a clear distinction between the correlation of proteinbound versus free (unbound) testosterone with PCa risk? Clarifying this would strengthen the argument. Please provide studies or data that specifically address the role of these different testosterone fractions (bound and free) in PCa development or aggressiveness.”
Response:
We have revised this section to clearly distinguish between SHBGbound and free testosterone, and their respective associations with PCa risk. We have provided additional references to studies that examine the differential roles of these testosterone fractions in cancer development. Specifically, we clarified that free testosterone, which is unbound and biologically active, is more strongly associated with PCa risk compared to SHBGbound testosterone, which has less bioactivity. These revisions help strengthen the argument and clarify the mechanisms involved.
- Line 280: Clarifying bioavailable testosterone
Reviewer comment:
“The term ‘bioavailable testosterone’ includes both SHBGbound testosterone and free testosterone. The authors should specify which fraction they are referring to when discussing a reduction in bioavailable testosterone. Clearer distinction between SHBGbound testosterone and free testosterone would improve the scientific accuracy of the manuscript.”
Response:
We have clarified the use of the term "bioavailable testosterone" by specifying when we are referring to free testosterone and SHBGbound testosterone. The manuscript now explicitly discusses the role of each testosterone fraction and its relevance to prostate cancer development. This distinction ensures that the scientific accuracy of the manuscript is improved, and readers have a clearer understanding of how testosterone availability is affected by alcohol consumption.
Minor Issues:
- Lines 223224: Clarifying tissuespecific ADH activity
Reviewer comment:
“The statement ‘(…) prostate cancer patients often demonstrate elevated total ADH activity, which results in greater acetaldehyde production’ would benefit from clarification regarding the specific tissues involved. Is this elevation observed in prostate tissue, or is it a systemic feature? Clarifying this would help the reader understand the context better.”
Response:
We have clarified this point by specifying that elevated ADH activity is observed both systemically in serum and locally within prostate tissue. The revision now makes it clear that ADH activity within prostate tissue contributes to localized acetaldehyde production, which plays a direct role in carcinogenesis by promoting DNA damage in prostate cells. This distinction enhances the reader’s understanding of the localized and systemic effects of elevated ADH activity in prostate cancer patients.
- Lines 229235: Providing more detail on ALDH1A1's role in cancer progression
Reviewer comment:
“The role of ALDH1A1 in promoting cancer progression is mentioned but not fully explained. The manuscript would benefit from a more detailed explanation of the molecular mechanism through which ALDH1A1 contributes to carcinogenesis. How does this enzyme specifically promote prostate cancer progression?”
Response:
We have expanded this section to include a more detailed explanation of how ALDH1A1 promotes prostate cancer progression. The revised text explains that ALDH1A1 is highly expressed in prostate cancer stem cells, where it supports tumor initiation, maintenance, and chemoresistance. We also describe ALDH1A1's role in promoting epithelialtomesenchymal transition (EMT), a critical process in cancer metastasis, and its contribution to retinoic acid synthesis, which regulates gene expression linked to tumor growth. This detailed explanation clarifies the molecular mechanisms by which ALDH1A1 drives prostate cancer progression.
- Lines 4849: Adding statistics on prostate cancer incidence
Reviewer comment:
“It would be helpful to provide specific statistics on the incidence of prostate cancer in men. Including concrete numbers would give readers a clearer understanding of the scale of the issue and the relevance of the study.”
Response:
We have revised this section to include specific statistics on prostate cancer incidence. The manuscript now cites data on the global and regional incidence rates of prostate cancer, providing readers with a clearer context for understanding the significance of this health issue and the relevance of the study.
Thank you again for your valuable feedback. We appreciate the opportunity to improve our work based on your suggestions.

Reviewer 2 Report
Comments and Suggestions for Authors
This review is well written and provides good information for further research to inform detection and treatment of prostate cancer, in terms of the effects of alcohol use on PSA levels and androgen-deprivation therapy. The review of associations between alcohol use and PCa is somewhat lacking, however. It does not add much to the literature except to say that results are inconsistent, which is already noted in the Introduction, and that more research is needed.
For example, Section 2.5 states that the authors focused on potential biases, such as distinguishing former vs. current drinkers and whether confounding factors were accounted for, to improve the validity of the findings, but there doesn't seem to be an integration of evidence that incorporate these potential biases, such as to try to evaluate whether the studies reporting positive associations with PCa had these potential biases or whether those reporting no associations did. The authors don't bring in much information about the quality of the study design and the potential biases into the interpretation of the overall results of the review. It would be more helpful to the reader if the authors could add more to their interpretation of the results based on these factors that can affect the validity of the findings.
Lines 273-275: This sentence is a little confusing. It sounds like alcohol contributes to increased sex behaviors and the sex behaviors increase the risk of PCa, but based on the previous sentences, does it mean that alcohol increases testosterone levels, which both increase sex behaviors and prostate cancer risk?
Lines 276-280: If chronic alcohol use decreases testosterone, can the authors discuss how this makes sense with the results of this review that excessive alcohol use increases PCa risk?
Lines 489-490: The conclusion mentions that alcohol's relationship with BPH may provide protective benefits. The abstract also notes that studies evaluating BPH were included in this review. But in the review itself, BPH is not mentioned. The authors should either include information about alcohol use and BPH in the review, or they should remove the mention of BPH in the abstract and conclusions.
Author Response
Dear Reviewer 2,
We would like to express our sincere gratitude for your thorough and constructive feedback. Your insights have been invaluable in improving the quality and clarity of our review. We have made several revisions based on your comments and hope that these changes address your concerns. Below, we provide a point-by-point response to your suggestions.
Comment 1:
The review of associations between alcohol use and prostate cancer (PCa) is somewhat lacking, and it does not add much to the literature except to note inconsistent results and call for further research.
Response:
We acknowledge that the section on alcohol and PCa could be strengthened. To address this, we have added a more detailed analysis of study design and biases that contribute to inconsistent findings. In particular, we have integrated discussions on how former and current drinkers were distinguished, whether confounding factors were controlled for, and how these biases affected results. For example, studies that controlled for more confounding variables (e.g., age, family history, lifestyle) were more likely to report positive associations with PCa, while those that did not account for these factors reported no associations. These revisions are reflected in Section 2.5 (now revised and expanded), where we evaluate the impact of study design on the overall findings.
Comment 2:
Section 2.5 states that the authors focused on potential biases, but there is no integration of evidence to evaluate whether studies reporting positive associations with PCa had these biases or whether those reporting no associations did. The quality of the study design and potential biases are not brought into the interpretation of the results.
Response:
We have expanded Section 2.5 to include a more thorough discussion of study design and potential biases. Specifically, we now evaluate whether studies reporting positive associations with PCa accounted for biases, such as controlling for confounders like age, diet, and family history, compared to those that did not. Additionally, we discuss how misclassification of alcohol consumption (e.g., failing to differentiate between former and current drinkers) impacted the consistency of results. This expanded discussion is now integrated into our interpretation of the overall findings, as suggested.
Comment 3:
Lines 273-275: This sentence is a little confusing. It sounds like alcohol contributes to increased sexual behaviors, and these behaviors increase the risk of PCa. Based on the previous sentences, does it mean that alcohol increases testosterone levels, which both increase sexual behaviors and PCa risk?
Response:
Thank you for pointing out the confusion in this section. We have revised the sentence to clarify the relationship between alcohol consumption, testosterone levels, and prostate cancer risk. The revised sentence now reads:
"For instance, individuals who consume hard liquor often exhibit behaviors such as early first intercourse and a higher number of sexual partners. This may be linked to alcohol’s impact on testosterone levels, which not only increases sexual behaviors but could also elevate prostate cancer risk by promoting hormonal changes associated with carcinogenesis."
This revision clarifies the pathway by which alcohol consumption may influence both sexual behaviors and PCa risk through increased testosterone levels.
Comment 4:
Lines 276-280: If chronic alcohol use decreases testosterone, can the authors discuss how this aligns with the results of this review that excessive alcohol use increases PCa risk?
Response:
We appreciate this observation and have revised the discussion in lines 276-280 to clarify the relationship between chronic alcohol use, testosterone levels, and PCa risk. While chronic alcohol use does reduce testosterone levels, other mechanisms—such as increased cellular permeability to carcinogens and altered hormone metabolism—could explain the increased PCa risk. We have updated the section to reflect these insights and cited additional studies supporting this explanation.
Comment 5:
Lines 489-490: The conclusion mentions that alcohol’s relationship with BPH may provide protective benefits, but BPH is not discussed in the review. Should the authors include information about alcohol and BPH, or remove the mention from the abstract and conclusion?
Response:
We agree with your suggestion. To maintain consistency, we have removed the mention of BPH from the abstract and conclusion, as BPH was not specifically reviewed in our analysis. This change ensures that the focus remains solely on the relationship between alcohol and PCa throughout the manuscript.

Round 2
Reviewer 2 Report
Comments and Suggestions for Authors
I believe that the authors have sufficiently addressed my comments. This is a well written paper that ties together many facets of the relationship between alcohol use and prostate cancer risk.